# Availability of mobile phones for discharge follow-up of pediatric Emergency Department patients in western Kenya

Darlene R. House[1,2], Philip Cheptinga[2,3] and Daniel E. Rusyniak[1]

[1] Department of Emergency Medicine, Indiana University School of Medicine, Indianapolis, IN, USA
[2] Academic Model Providing Access to Healthcare (AMPATH), Eldoret, Kenya
[3] Department of Child Health and Paediatrics, Moi University School of Medicine, Eldoret, Kenya

## ABSTRACT

**Objective.** Mobile phones have been successfully used for Emergency Department (ED) patient follow-up in developed countries. Mobile phones are widely available in developing countries and may offer a similar potential for follow-up and continued care of ED patients in low and middle-income countries. The goal of this study was to determine the percentage of families with mobile phones presenting to a pediatric ED in western Kenya and rate of response to a follow-up phone call after discharge.

**Methods.** A prospective, cross-sectional observational study of children presenting to the emergency department of a government referral hospital in Eldoret, Kenya was performed. Documentation of mobile phone access, including phone number, was recorded. If families had access, consent was obtained and families were contacted 7 days after discharge for follow-up.

**Results.** Of 788 families, 704 (89.3%) had mobile phone access. Of those families discharged from the ED, successful follow-up was made in 83.6% of cases.

**Conclusions.** Mobile phones are an available technology for follow-up of patients discharged from a pediatric emergency department in resource-limited western Kenya.

## INTRODUCTION

In developed countries, mobile phones have been successfully used for Emergency Department (ED) patient follow-up. They have been used to continue discharge education, to evaluate treatment compliance, to identify and intervene when patients are clinically worsening, to assess patient satisfaction, and to improve upon patient experiences (*Arora et al., 2014*; *Guss, Leland & Castillo, 2013*; *Jones et al., 1988*; *Lee, 2004*; *Patel & Vinson, 2013*). For example, *Arora et al. (2014)* found that patients who received mobile phone follow-up had improved medication compliance and disease management. In another study, Chande and Exum found that ED patients who received a follow-up phone call were more likely to follow-up with their primary care physicians compared to those who received no follow-up phone call (*Chande & Exum, 1994*). Additionally, *Jones et al. (1988)* showed the utility of a

Corresponding author
Darlene R. House,
dhouse@iupui.edu

telephone follow-up system in providing improved clarification of discharge instructions and in intervening with patients who are doing poorly. Overall, the use of phones for patient follow-up have been shown to improve the care of patients discharged from the ED (*Arora et al., 2014*; *Chande & Exum, 1994*; *Jones et al., 1988*).

Mobile phones are also widely available in developing countries with more than 75% of the world's population having access to a mobile phone (*World Bank Group, 2012*). Given the lack of primary care follow-up and resource constraints, mobile phones offer a similar potential for follow-up of ED patients in low and middle-income countries (LMIC). Despite this, there have been no studies to date showing their availability for ED follow-up in LMIC settings.

We designed a study to determine the percentage of ED patient families with mobile phones presenting to a pediatric ED in resource-limited western Kenya. Furthermore, we determined the rate of response to a follow-up phone call seven days after ED discharge.

## MATERIALS AND METHODS

### Study design

A prospective, cross-sectional observational study of children presenting to Moi Teaching and Referral Hospital in Eldoret, Kenya was performed between the months of February and April 2013. The study was approved by both Indiana University Institutional Review Board (Approval IRB #1301010311) and the Institutional Research and Ethics Committee at Moi Teaching and Referral Hospital.

### Study setting

Eldoret is a town of approximately 220,000 people located in western Kenya and serves as the administrative center of Uasin Gishu District of Rift Valley Province. Eldoret is home to Moi University School of Medicine and Moi Teaching and Referral Hospital (MTRH). MTRH is a government referral hospital in western Kenya that serves a catchment population of 13 million people. All sick children that present to the referral hospital are evaluated and treated in the MTRH Sick Child Clinic. The Sick Child Clinic serves as the pediatric emergency department for the hospital, evaluating all acute pediatric medical and trauma patients. The department evaluates and treats approximately 100 children a day. Patients are either seen by a clinical officer (equivalent to an advanced practitioner) or a medical officer (physician). MTRH is also the base for a partnership between the United States Agency for International Development (USAID) and the Academic Model Providing Access to Healthcare (AMPATH), an organization that provides HIV care, primary health care, and chronic disease management.

### Study population

All families with children presenting and subsequently discharged from the Sick Child Clinic during the study were eligible. Patients admitted were excluded from the study as the remainder of their care is followed by the inpatient pediatric team while discharged patients have no clear follow-up in this setting.

**Table 1  Patient demographics.**

| | |
|---|---|
| Male | 53.1% |
| Female | 46.9% |
| Age | |
| <1 mo | 4.2% |
| 1 mo–12 mo | 24.0% |
| 1 y–5 y | 40.6% |
| >5 y | 31.2% |

## Study protocol

Convenience sampling was performed during daytime ED hours. Documentation of access to mobile phones, including phone number, was recorded during registration of patients presenting to the Sick Child Clinic. If families had access to a mobile phone, consent was obtained to contact them for follow-up at seven days after being discharged from the ED. Families of patients who were admitted from the Sick Child Clinic were not contacted for follow-up. At seven days from discharge, an attempt to contact the patient's family was made. To provide further construct validity to the use of mobile phones for follow-up of ED patients, a clinical officer made these calls using a script to obtain the following information: how their child was doing, any return visits within the seven days from discharge, comprehension of ED diagnosis and plan, compliance with medications, and any complications. Patients who had not shown any improvement or were doing worse were instructed to return for further evaluation. If the phone call was not answered, a repeat call was made each subsequent day. If after three attempts there was no answer, the follow-up was classified as unsuccessful.

## Statistical analysis

Descriptive statistics regarding families' access to mobile phones, successful follow-up, and status upon follow-up were calculated.

## RESULTS

Of the 1,168 patients seen in the Sick Child Clinic during the study, 854 patients were seen during sampling hours. Of these patients, 788 families were approached for the study. A total of 704 (89.3%) families had mobile phone access; of those, 659 families were eligible for the study (see Fig. 1). Successful follow-up was made in 551 (83.6%) of these cases.

Most patients seen were under five years of age (see Table 1). Also, the majority of patients had either a respiratory or gastrointestinal illness (see Table 2).

In families successfully contacted, 490 patients were reported to be doing better, 50 patients were reported to be doing the same, and 11 patients were reported to be worse. Of the patients who were doing better at the time of follow-up, 20 had been seen again since their initial visit, four had been admitted and discharged. Of the 50 caregivers who reported their child was doing the same, nine patients had been seen again since their initial visit and five of these patients were admitted. The remaining 41 patients were

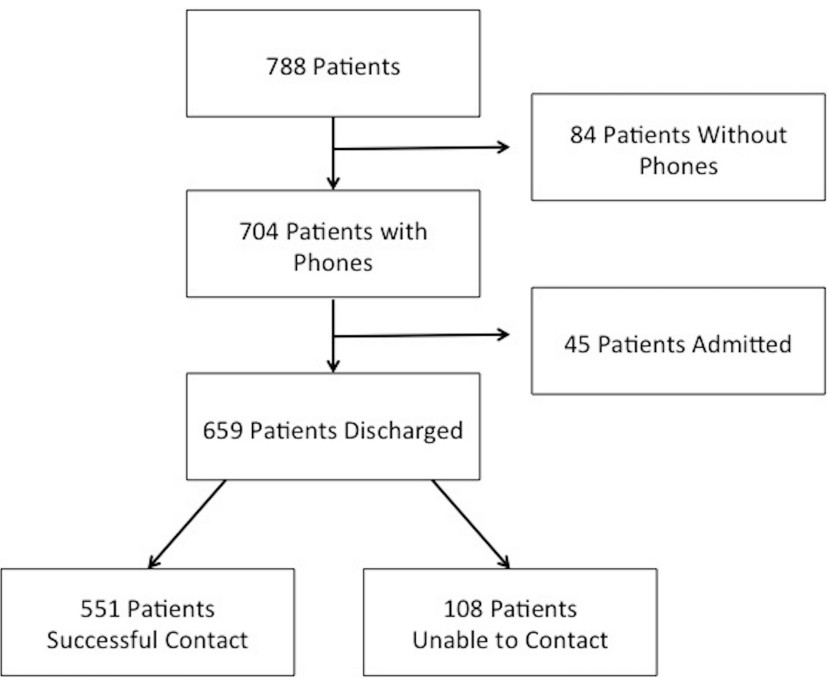

**Figure 1  Patient enrollment and follow-up.**

| Table 2  Patient diagnoses. | |
| --- | --- |
| Respiratory tract infections | 30% |
| Vomiting/diarrhea | 25% |
| Injuries | 12% |
| Seizure | 8% |
| Febrile illness | 8% |
| Rashes | 6% |
| Urinary tract infection | 6% |
| Other | 5% |

instructed to return for further evaluation. Of the patients who had worsened, seven patients had been seen again and subsequently admitted. Two of those patients died. The other four patients who reported to be worse were told to return for further evaluation.

Regarding patient instructions, 488 parents reported understanding their discharge diagnosis and instructions and 63 asked for further clarification. Only 23 patients reported not being compliant with their medications.

## Limitations

The study was done in a referral center that requires payment for care of children. This potentially biases our population toward families having the resources to own a mobile phone. Further studies evaluating the availability of mobile phones in more rural areas in resource-limited settings should be conducted. Another limitation is potential bias with

convenience sampling. While nearly all families that were seen during the sampling hours were approached for the study, families that came at night may potentially have less or more access to mobile phones. Further study would need to be done that could include families presenting at all hours to assess if any differences exist. Also, another limitation is that data only represents what families self-reported; this offers the potential for recall bias and false reporting. Although the data is self-reported, the study provides evidence that families were accessible and receptive to receive phone follow-up.

## DISCUSSION

This study provides the first evidence suggesting that mobile phones may be an effective tool for following up patients discharged from an ED in a resource-limited country. Nearly all families presenting to the ED had access to mobile phones and the majority could be reached by phone to discuss their child's condition.

Our study found 89% of our ED families had access to a mobile phone. Similarly, a previous study done in Kenya also found 85% of Kenyans surveyed had access to a mobile phone; however, phone ownership diminished with mean monthly income (*Wesolowski et al., 2012*). This is just slightly lower than found in the US, where 95% of ED patients have access to mobile phones (*Ranney et al., 2012*). Additionally, we found that we could successfully follow-up with patients 83.6% of the time. This is better than most follow-up rates found in the US that varied from 68–73% (*Menchine et al., 2013*; *Suffoletto et al., 2012*). However, one study done in the US by *Jones et al. (1988)* had a similar follow-up rate of 81% with three phone call attempts. Therefore, with similar rates to the US where phones have been successfully implemented for ED follow-up care, the availability and high response rates to follow-up phone calls in this resource-limited setting offers tremendous opportunity to provide ongoing care to ED patients.

Follow-up is especially important in developing countries where morbidity and mortality are high, resources are limited, and primary care is almost nonexistent. In our study, an appreciable number of patient families were either unsure of discharge instructions, not compliant with medications, or clinically unimproved. Mobile phone follow-up, therefore, provides the health care provider another opportunity to educate families and provide further medical advice regarding evaluation and treatment. Also with an economically disadvantaged patient population, patients may delay seeking care given the costs of transportation and medical visits. This invites the use of low-cost means to obtain patient follow-up. Using mobile phones is a potential solution. Since phone calls made to patient families are free to the patients, this provides an additional medical service to families without any additional financial burden. Telephone follow-up provides a free, for the patient, means to provide ongoing care and education and could potentially prevent revisits, decrease travel and associated costs, and provide the opportunity to intervene if patients are deteriorating. Although follow-up phone calls add expense to the ED related to phone usage and staffing, the cost may be offset by improved patient outcomes and improved patient satisfaction. Another advantage of mobile phones, as this

study demonstrates, is they can be used to collect data for quality improvement or research purposes.

In a similar study, *Maslowsky et al. (2012)* showed that patients discharged after hospital admission in Ecuador could be reached by mobile phone for further post-hospitalization medical advice and facilitation of further care. Other studies done in resource-limited settings have implemented text messaging for evaluating disease management and medication compliance among patients being treated for both communicable and non-communicable diseases (*Deglise, Suggs & Odermatt, 2012*; *Lester et al., 2010*; *Pop-Eleches et al., 2011*). While we did not implement text messaging due to cost to families to text back, all the phones of the families in our study had texting capabilities. This could be another modality for ED patient follow-up in resource-limited settings. Together with our study, these studies highlight the broad applicability of mobile phones for both ED and hospital discharge follow-ups.

Accessibility of mobile phones in resource-limited settings provides the groundwork for further research and clinical opportunities. With demonstrated availability of mobile phones and responsiveness upon follow-up in this ED, next steps would include replicating ED discharge follow-ups in other developing countries as well as more rural areas to determine broader application. Next steps would also include randomized studies assessing outcomes and effectiveness of mobile phone use for follow-up compared to standard discharge instructions. Other applications could include development of a pre-hospital call system to guide home care, offering cost-effective medical advice, disease management and treatment, as well as triaging patients to appropriate levels of care. Developing pre-hospital and ED discharge follow-up systems that utilize mobile phones has great potential to improve emergency care of patients in resource-limited settings.

## CONCLUSION

Mobile phones are an available technology for follow-up of patients discharged from a pediatric emergency department in resource-limited western Kenya.

### Funding

The authors declare there was no funding for this work.

### Competing Interests

The authors declare there are no competing interests.

### Author Contributions

- Darlene R. House conceived and designed the experiments, performed the experiments, analyzed the data, contributed reagents/materials/analysis tools, wrote the paper, prepared figures and/or tables, reviewed drafts of the paper.

- Philip Cheptinga conceived and designed the experiments, performed the experiments, analyzed the data, contributed reagents/materials/analysis tools, wrote the paper, reviewed drafts of the paper.
- Daniel E. Rusyniak conceived and designed the experiments, analyzed the data, contributed reagents/materials/analysis tools, wrote the paper, reviewed drafts of the paper.

### Human Ethics

The following information was supplied relating to ethical approvals (i.e., approving body and any reference numbers):

Indiana University IRB (Approval IRB # 1301010311) and Institutional Ethics and Review Committee at Moi Teaching and Referral Hospital in Eldoret, Kenya (Approval #000936).

### Supplemental Information

Supplemental information for this article can be found online at http://dx.doi.org/10.7717/peerj.790#supplemental-information.

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
