# Peer review of "Availability of mobile phones for discharge follow-up of pediatric Emergency Department patients in western Kenya"

_PeerJ, doi:10.7717/peerj.790_

## Round 0.1 · original submission · Major Revisions

Please address the comments raised by the reviewers. Its important to describe criteria for participation and exclusion. Ensure that your results reflect the aims and objectives of the study.

·

Basic reporting

The claims on line 37, that phones can improve ED care - can you reference and define this further. Researchers and the market are looking for phone based interventions that can improve care - but a definition of this improvement is important - outcomes for discharged patients - mortality, morbidity, return to medical home, decrease 72 hour returns, patient satisfaction, etc. This statement could be misleading.

Experimental design

Line 51 - Can you identify the institutional IRB approving the study?
Can you clarify if the study was conducted in the ED or the outpatient clinic? Can you define what a clinical office is ? MD, RN, CHW or other ?

Validity of the findings

Last sentence of limitations - line 112 might be better in the discussion.

Additional comments

this study helps establish the reach of mobile phones in this population. The high follow up rate is encouraging and is useful information for designing further studies and interventions.

·

Basic reporting

Introduction:

You say there is no data for mobile phone availability data for ED follow-up in LMIC settings, what about general population mobile phone ownership in LMIC's? This data is available.

Methods

1) Was there an exclusion criteria? I presume it was the patients admitted from Sick Child Clinic? Please be clear.
2) Why were families of patients who were admitted from Sick Child Clinic not contacted for follow-up?
3) I am unsure as to who you are collecting this data from, at one point the authors say it is from the ED patients (line 44) at another it is from the families of the patients (line 22)?

Results

1) The authors say that ' there was no difference in mobile phone access or successful follow-up based on age or illness'. Based on what?There are no statistics reported.
2) I am unsure as to the relevance of reporting the condition of patients at follow-up? Please clarify, the objective of the study is to determine mobile phone availability?

Table

Check guidelines on rounding numbers

Discussion

What would a further study assessing outcomes and effectiveness of mobile phone use for follow-up look like? More details maybe?

There are grammatical issues:
1) Acronym for 'ED' is not defined.
2) There are full stops where there shouldn't be (line 36), and no full stops where there should be (Line 37). I assume this is because of a reference manager.
3) No capital letter after a colon (line 24).
4) Numbers under 10 are written as a word.

Please check all grammar, this is not exhaustive.

Experimental design

fine

Validity of the findings

Please see basic reporting section

Additional comments

Overall this is a necessary, timely and informative piece of work. There are a few issues around clarity of reporting and grammar which need to be checked. I will recommend for publication once these have been addressed.

---

## Round 0.2 · Minor Revisions

Thank you for the resubmitted article in which you have addressed most of the concerns of the reviewers. However before I can recommend this for publication you need to further develop your background / introduction as at present it does not adequately provide a review of the relevant literature and explain how your study can be placed in the wider context. Your discussion section would also befit from a more extensive reflection upon your findings, their limitations and how this work can be further refined in future studies.

---

## Round 0.3 · accepted · Accept

Thank you for your revised manuscript. This now includes the additional material suggested. I am happy to recommend publication.